

# Quality risk management for microbial control in membrane-based water for injection production using fuzzy-failure mode and effects analysis

Luoyin Zhu and Yi Liang

School of International Pharmaceutical Business, China Pharmaceutical University, Nanjing, China

## ABSTRACT

Microbial proliferation presents a significant challenge in membrane-based water for injection (WFI) production, particularly in systems with storage and ambient distribution, commonly refered to as cold WFI production. A comprehensive microbial risk assessment of membrane-based WFI systems was performed by employing Fuzzy-Failure Mode and Effects Analysis (Fuzzy-FMEA) to evaluate the potential microbial risks. Failure modes were identified and prioritized based on the Risk Priority Number (RPN), with appropriate preventive measures recommended to control failure modes that could increase the microbial load and mitigate their impact. Key hazards were identified including fouling of ultrafiltration (UF) membranes, insufficient sealing of heat exchangers, leakage in reverse osmosis (RO) membranes, and ineffective vent filters unable to remove airborn microorganism. Based on Fuzzy-FMEA results, suggestions for optimization were proposed to improve microbial control in membrane-based WFI systems in the pharmaceutical industry.

# INTRODUCTION

Pathogenic bacteria and endotoxins in water for injection (WFI) pose significant health risks when entering the human body, leading to conditions such as fever, diarrhea, vomiting, and acute respiratory illnesses (*Janik et al., 2020*; *Rasuli et al., 2022*; *Sattar et al., 2022*). To mitigate these microbial hazards, stringent microbial and endotoxin limits are essential for WFI. Traditionally, WFI production relied on distillation due to its reliable delivery of water, meeting strict quality standards (*ISPE, 2022*). However, recent advancements in membrane-based processes, alongside the development of online instrumentation and reduced steam demand, have led to increasing acceptance of membrane-based WFI systems (*Cataldo et al., 2020*; *Herold, 2021*). Given the high-risk nature of microbial and endotoxin contamination in WFI systems, especially in membrane-based setups, effective risk management is crucial to safeguard public health. Membrane-based systems lack the continuous thermal lethality provided by distillation, making them particularly susceptible to microbial proliferation. Consequently even with

Corresponding author
Yi Liang, ly606@sohu.com

these technological advancements, microbial contamination remains a primary risk, as membrane-based WFI lacks the continuous heat lethality characteristic of distillation (*ISPE, 2022*).

Membrane-based WFI systems typically consist of three subsystems: pretreatment, production, and storage/distribution. A critical need exists for a comprehensive risk assessment of microbial growth in these systems, especially given the absence of constant heat lethality. Risk, defined as the combination of likelihood, frequency, and severity of harm (*Aven, 2010*; *ICH Expert Working Group, 2023*), cannot be fully eliminated, while can be mitigated (*Aven, 2010*). Various studies have examined risk reduction strategies for microbial contamination in pharmaceutical water systems, and methods, such as reverse osmosis (RO) and electrodeionization (EDI) revealed promising results (*Chen et al., 2021*; *Sackstein, 2017*). Advanced techniques in surface modification have also been developed to reduce bacterial adhesion and biofilm formation, as exemplified by *Filice et al.*'s *(2022)* smart surface approach targeting *Legionella pneumophila*. Additionally, several risk assessment methods, such as Hazard Analysis Critical Control Point (HACCP), fault tree analysis, and Shewhart control charts, were applied to optimize the design and operational integrity of water systems (*Beauchamp, Lence & Bouchard, 2010*; *da Costa et al., 2022*; *Eissa, 2015*; *Uriadnikova et al., 2022*).

While Failure Mode and Effects Analysis (FMEA) has been widely applied in pharmaceutical water systems, it is less frequently utilized for WFI-specific applications, and studies have mainly concentrated on purified water systems (*Rimantho et al., 2017*; *Sahu et al., 2016*). However, the Fuzzy-FMEA, a recent advancement that integrates fuzzy inference for enhanced risk prioritization, has proven effective in water treatment risk analysis (*Huang et al., 2020*). For instance, *Alizadeh et al. (2022)* used the Fuzzy-FMEA for evaluating risks in municipal wastewater plants, while *Haider et al. (2021)* employed a similar approach to assess water quality risks.

Despite the importance of microbial risk management in pharmaceutical water systems, existing studies have concentrated primarily on traditional FMEA approaches, which are limited in their ability to accurately prioritize risks and account for complex interactions between failure modes. Although some studies have explored FMEA enhancements, such as fuzzy logic in various water treatment contexts, none have applied Fuzzy-FMEA specifically to membrane-based WFI systems.

This study addressed this gap by developing a microbial risk profile for membrane-based WFI systems, concentrating on storage and ambient distribution. As the first research to incorporate fuzzy logic theory into FMEA for evaluating microbial risk in membrane-based WFI, this research provided a valuable basis for stakeholders to make informed, risk-based decisions to enhance system robustness. The remainder of this article is structured as follows. "Methodology" details the risk analysis model and techniques employed in this study. "Application" presents a case study on microbial risk assessment in membrane-based WFI systems, comparing results from the proposed the Fuzzy-FMEA approach with traditional methods and discussing key findings. The final section concludes with insights on the study's contributions and practical implications for quality risk management in pharmaceutical water systems.

## METHODOLOGY

This study aimed to propose a hybrid approach, integrating fuzzy logic with FMEA to assess microbial risk in membrane-based WFI systems. The methodology developed in this research is detailed as follows.

### The cause-effect fishbone diagram

Identifying potential failure modes is a fundamental step in FMEA (*Xu et al., 2020*). The fishbone diagram, also known as the Ishikawa or cause-and-effect diagram, is a widely used tool for root cause analysis (*Ito et al., 2022*). Originally developed by Ishikawa in 1990, this quality control tool was designed to identify factors contributing to an overall effect and mitigate product quality defects (*Coccia, 2020*). In the present study, the fishbone diagram was employed to identify and analyze the root causes of microbial issues in membrane-based WFI systems.

### Conventional FMEA

FMEA is a structured approach used to detect and mitigate potential failures or defects within processes, products, or systems (*Sharma & Srivastava, 2018*). Originally developed in the 1940s for the aerospace and military sectors to enhance system reliability and reduce the chances of failure (*Demirkaya, 2022*), FMEA has since been adopted across diverse industries, including automotive (*Petrescu, Cazacu & Petrescu, 2019*), healthcare (*Abbassi, Brahim & Ouahchi, 2023*), and manufacturing (*Demirkaya, 2022*), to optimize product and process design, improve reliability, and lower failure-related costs.

After identifying potential failure modes in a system, each was prioritized based on its risk priority number (RPN), which calculated by multiplying three risk parameters: severity ($S$), occurrence ($O$), and detection ($D$). Once assigned, these ratings yielded an RPN value through Eq. (1), where higher values indicated a higher priority for addressing the specific failure mode:

$$RPN = O \times S \times D \tag{1}$$

The rating scale (Tables 1–4) assigns scores for severity, occurrence, and detection for each failure mode (FM) based on pharmaceutical experts' insights.

While FMEA is a valuable tool for risk assessment and mitigation, it has certain limitations (*Dai et al., 2011*; *Selvan et al., 2013*; *Septiyana, 2021*; *Sharma, Kumar & Kumar, 2005*):

i) FMEA assigns equal weight to severity, occurrence, and detection, which may not accurately reflect their relative importance in specific situations. For instance, a failure mode with high severity while low occurrence may be more critical than one with low severity and high occurrence.

ii) Certain combinations of severity, occurrence, and detection scores can result in identical RPN values, even when the underlying risk profiles differ significantly.

**Table 1 Severity assignment criteria (S).**

| Linguistic variables | Score | Descriptions |
|---|---|---|
| Almost none | 1 | The impact of microbial level in the produced water is negligible. In daily operations, such impacts do not affect product quality or the production process in a detectable manner, and no special measures are required. |
| Low | 2 | The microbial level has slightly increased, but it remains well below the safety threshold. It will not impact product quality in the short term, and routine water treatment processes are sufficient to manage it. |
| | 3 | The increase in microbial level is detectable, but its impact on product quality remains minimal. Minor adjustments to the water treatment process may be needed, but it will not cause production downtime. |
| Medium | 4 | The microbial count has increased to a level that requires attention. While it will not have a significant impact on the product in the short term, monitoring and appropriate adjustments are necessary to prevent future risks. |
| | 5 | The increase in microbial level is beginning to have a mild impact on product quality. Moderate modifications to the treatment process and increased monitoring frequency may be required to ensure that the issue does not worsen. |
| | 6 | The microbial level in the produced water has a more noticeable impact, which could affect the quality of specific batches if unaddressed. Immediate actions are required to avoid further issues. |
| High | 7 | The increase in microbial level has significantly affected the quality of several batches of products. Urgent measures are needed, possibly including halting the production line for thorough cleaning and disinfection. |
| | 8 | The microbial level in the produced water has a severe impact, affecting multiple continuous production batches. Major intervention measures are necessary, potentially involving significant adjustments to the production process. |
| Very high | 9 | The increase in microbial level has had a profound impact on the entire production line, leading to serious production disruptions. Substantial resources are needed for problem identification and resolution, along with prolonged cleaning and validation processes. |
| | 10 | The microbial level in the produced water has a severe impact; when used in product manufacturing, it can affect all subsequent batches in a continuous production for weeks or months. Addressing this issue requires considerable cost and time investment, possibly including innovative technological solutions. |

**Table 2 Occurrence assignment criteria (O).**

| Linguistic variables | Score | Descriptions |
|---|---|---|
| Almost none | 1 | The probability of the event occurring is nearly zero. Under normal circumstances, it is almost never encountered, only possible under extremely special or abnormal conditions. |
| Low | 2 | The event occurs occasionally, but it is unlikely in most situations. It may only be triggered under very specific conditions. |
| | 3 | The frequency of the event is low; most of the time, issues will not be encountered. It only appears under certain specific circumstances. |
| Medium | 4 | The event occurs occasionally but is not the norm. Under specific conditions or environments, its frequency might increase. |
| | 5 | The event has a definite frequency of occurrence; it is neither rare nor common. It may be encountered occasionally in daily operations. |
| | 6 | The frequency of the event is starting to become somewhat frequent, requiring constant vigilance. Under specific conditions, the likelihood of occurrence is higher. |
| High | 7 | The occurrence of the event has become a normal part of expectations, with several encounters expected within a certain period. Measures need to be taken to reduce its occurrence. |
| | 8 | The frequency of the event is very high, almost a part of daily activities. Its occurrence must be considered in operations to prevent and prepare. |
| Very high | 9 | The event is almost inevitable and occurs very frequently. Continuous and effective measures are needed to manage its impact. |
| | 10 | The frequency of the event is extremely high, nearly a constant condition. It significantly impacts daily operations, necessitating specially designed processes and contingency plans to address it. |

**Table 3 Comprehensive worksheet of membrane-based WFI system.**

| Linguistic variables | Score | Descriptions |
|---|---|---|
| Certain | 1 | The current method can precisely identify sources of hazards, with reliable and accurate detection methods available. This means that hazards can be quickly identified and located, allowing for appropriate measures to be taken. |
| High | 2 | The identification and detection of hazards are very effective, and although there are rare cases of errors, accurate results are provided in most situations. |
| | 3 | Reliable methods are available to identify most hazards, although additional effort or technology may be needed under certain specific conditions to ensure accuracy. |
| Medium | 4 | There are some effective detection methods available, but in some cases, it may be difficult to accurately identify hazards. Combining multiple methods may improve accuracy. |
| | 5 | While there are methods available to identify hazards, the reliability and accuracy of these methods are somewhat limited, requiring more time and resources for confirmation. |
| | 6 | The methods for detecting and identifying hazards have significant limitations, often requiring specialized knowledge and technology for assistance, and results are not always precise. |
| Low | 7 | There are few effective methods for accurately identifying hazards, often requiring complex processes and specialized technical support, with substantial uncertainty. |
| | 8 | There are almost no direct effective methods for identifying hazards; reliance on indirect signs and professional analysis is necessary, often facing a high risk of failure. |
| Very low | 9 | Identifying hazards is extremely difficult; most existing methods are ineffective, and detection can only be attempted through highly specialized and customized technology, with a low success rate. |
| | 10 | There are no effective methods available for identifying hazards. In this case, the detection and identification of hazard sources face great challenges, requiring new technological breakthroughs to solve. |

**Table 4 Definition of RPN.**

| Linguistic variables | Score | Descriptions |
|---|---|---|
| None | 1 | The severity is negligible, the frequency of occurrence is extremely low, and it is easy to detect. This level of hazard source is unlikely to impact the system and can be easily managed through routine means. |
| Very low | 2 | The severity is low, frequency of occurrence is not high, and identification and management are relatively easy. Such risks can usually be controlled through standard operating procedures and routine monitoring. |
| Low | 3 | The severity, frequency of occurrence, and detection difficulty are all below average. Some preventative measures and monitoring are needed, but they generally do not have a significant impact on overall safety and efficiency. |
| High low | 4 | At least one of the factors—severity, occurrence, or detection—is at a moderate level. This requires targeted management strategies to mitigate potential impacts. |
| Low medium | 5 | At least one factor (severity, occurrence, or detection) scores high, indicating that the hazard could lead to significant adverse consequences. Specialized risk management measures and emergency response plans are needed. |
| Medium | 6 | At least two of the factors—severity, occurrence, and detection—are rated high, indicating a serious safety or operational issue. This must be prioritized and addressed to avoid major impacts. |
| High medium | 7 | At least two of the factors—severity, occurrence, and detection—score high, indicating that the potential issues could cause significant or irreversible impacts. Immediate action and appropriate resource allocation are required to mitigate the risks. |
| Low high | 8 | All three factors—severity, occurrence, and detection—score high, showing that the potential impact of the hazard is great, the likelihood of occurrence is high, and it is difficult to detect. Such risks require urgent and comprehensive control measures and high-level attention. |
| High | 9 | This represents nearly the most severe level of risk in all cases, implying that the hazard could lead to catastrophic consequences, its occurrence is almost inevitable, and it is very difficult to detect and manage through conventional methods. |
| Very high | 10 | Scoring high in severity, occurrence, and detection, this represents the highest level of risk in extreme scenarios. Such risks could lead to fatal consequences, significant economic losses, or long-term adverse effects, necessitating the immediate implementation of the strictest control measures. |

iii) The RPN values are limited by the scale ranges for severity, occurrence, and detection, leading to clustering of values (*e.g.*, RPN 120 appears 24 times, while values, such as 1, 123, and 1,000 appear only once).

iv) FMEA calculates RPN for each failure mode independently, overlooking any interactions or dependencies between failure modes.

## Fuzzy-FMEA approach

To address the limitations of RPN values from conventional FMEA, this study applied fuzzy logic to FMEA, generating a fuzzy-RPN (fRPN) that overcomes these limitations and enhances the interpretability of the findings.

The Fuzzy-FMEA approach enhances traditional FMEA by incorporating fuzzy logic, which helps address common limitations found in conventional FMEA, including equal weighting of severity, occurrence, and detection parameters and the lack of accounting for dependencies among failure modes. By assigning flexible, linguistic values to risk parameters, the Fuzzy-FMEA provides a more nuanced assessment that can reduce the clustering of RPNs and avoid redundancy. Additionally, using fuzzy sets accommodates the subjectivity and variability in expert opinions, which often arise in risk evaluations in pharmaceutical water systems. However, the Fuzzy-FMEA requires a high level of expertise to define appropriate fuzzy sets and membership functions, which can make it resource-intensive compared to traditional FMEA. Moreover, the method relies on fuzzy rule bases, which may limit its accuracy if not calibrated correctly through extensive expert input.

### Fuzzy logic

In several real-world decision-making scenarios, key information sources include expert judgments expressed in natural language and sensory data modeled mathematically (*Bozanic et al., 2021*). Because each type of information is partial, an integrative approach is essential to translate human expertise into mathematical terms. Traditional binary logic is inadequate, as human reasoning mainly operates in degrees of truth rather than strict true or false values (*Goksu & Arslan, 2023*). Fuzzy logic was introduced to meet this need.

Fuzzy logic enables the design of a fuzzy inference system (FIS), mapping inputs to outputs using rules that reflect human reasoning (*Chicco, Spolaor & Nobile, 2023*). This approach encodes experiential knowledge in a format that computers can process. The three primary FIS models are Mamdani (*Vassilyev et al., 2020*), Sugeno (*de Campos Souza, 2020*), and Larsen (*Chaudhari et al., 2023*), and the Mamdani model is the most widely used for capturing expert knowledge (*Akyuz, Akgun & Celik, 2016*), which was adopted in this study.

Membership functions (MFs) convert net input values into fuzzy linguistic terms during fuzzification and defuzzification (*Jain & Sharma, 2020*). The degree of fuzziness in a fuzzy set is determined by MF values and shapes (*Vadiati et al., 2019*), typically represented by triangular, trapezoidal, or Gaussian distribution functions. This study utilized triangular and trapezoidal MFs.

The triangular MF, characterized by three parameters {a, b, c}, defines the degree ($\mu A(x)$) to which a net input value (x) belongs to a linguistic variable. Eq. (2) formulates the triangular MF:

$$\mu_A(x) = \begin{cases} 0, & x \leq a \\ \dfrac{x-a}{m-a}, & a < x \leq m \\ \dfrac{b-x}{b-m}, & m < x \leq b \\ 0, & x \geq b \end{cases} \quad \text{where a} \leq \text{m} \leq \text{b} \tag{2}$$

Trapezoidal MF is specified by four parameters {a, b, c, d} whose formula is presented in Eq. (3):

$$\mu_A(x) = \begin{cases} 0, & x < a \text{ or } x > d \\ \dfrac{x-a}{b-a}, & a < x \leq b \\ 1, & b \leq x \leq c \\ \dfrac{d-x}{d-c}, & c < x \leq b \end{cases} \quad \text{where a} \leq \text{b} \leq \text{c} \leq \text{d} \tag{3}$$

### *Fuzzy-FMEA approach*

A rule-based expert system approach is widely applied in fuzzy FMEA to address the limitations of conventional FMEA (*Akyuz, Akgun & Celik, 2016*; *Goksu & Arslan, 2023*). The foundation of this approach is a fuzzy rule base, a collection of IF-THEN rules that represent expert knowledge. In contrast to traditional methods relying on precise numerical values, fuzzy logic uses linguistic variables for input and output description. In the Fuzzy-FMEA applied in this study, severity (*S*), occurrence (*O*), and detection (*D*) serve as linguistic variables, with fuzzy sets defined as "remote," "low," "medium," "high," and "very high".

This study employed the Mamdani inference method to integrate the three risk-indexed parameters (*S, O, D*) nonlinearly through fuzzy IF-THEN rules, generating a more notable RPN. The Mamdani inference process involves five main steps: constructing the rule base, fuzzification, rule evaluation, rule aggregation, and defuzzification. The Fuzzy-FMEA approach for this study is outlined in Fig. 1 (*Ceylan, 2023*; *Goksu & Arslan, 2023*).

i. Rule Base Construction

The first step of a rule-based expert system is to collect fuzzy IF-THEN rules from domain experts. Fuzzy rules are statements that relate input variables to output variables in a linguistic form, typically expressed as "IF… THEN…" statements. A typical IF-THEN rule in the Fuzzy-FMEA can be expressed as follows (*Akyuz, Akgun & Celik, 2016*; *Goksu & Arslan, 2023*):

$$R_i : \text{If } o \text{ is } O_i, s \text{ is } S_i, \text{ and } d \text{ is } D_i, \text{ then, RPN is } R_i \ i = 1, 2, \ldots, K \tag{4}$$

where $R_i$ refers to the rule number, $K$ is the total number of rules, $O_i$, $S_i$, $D_i$, and $R_i$ are the
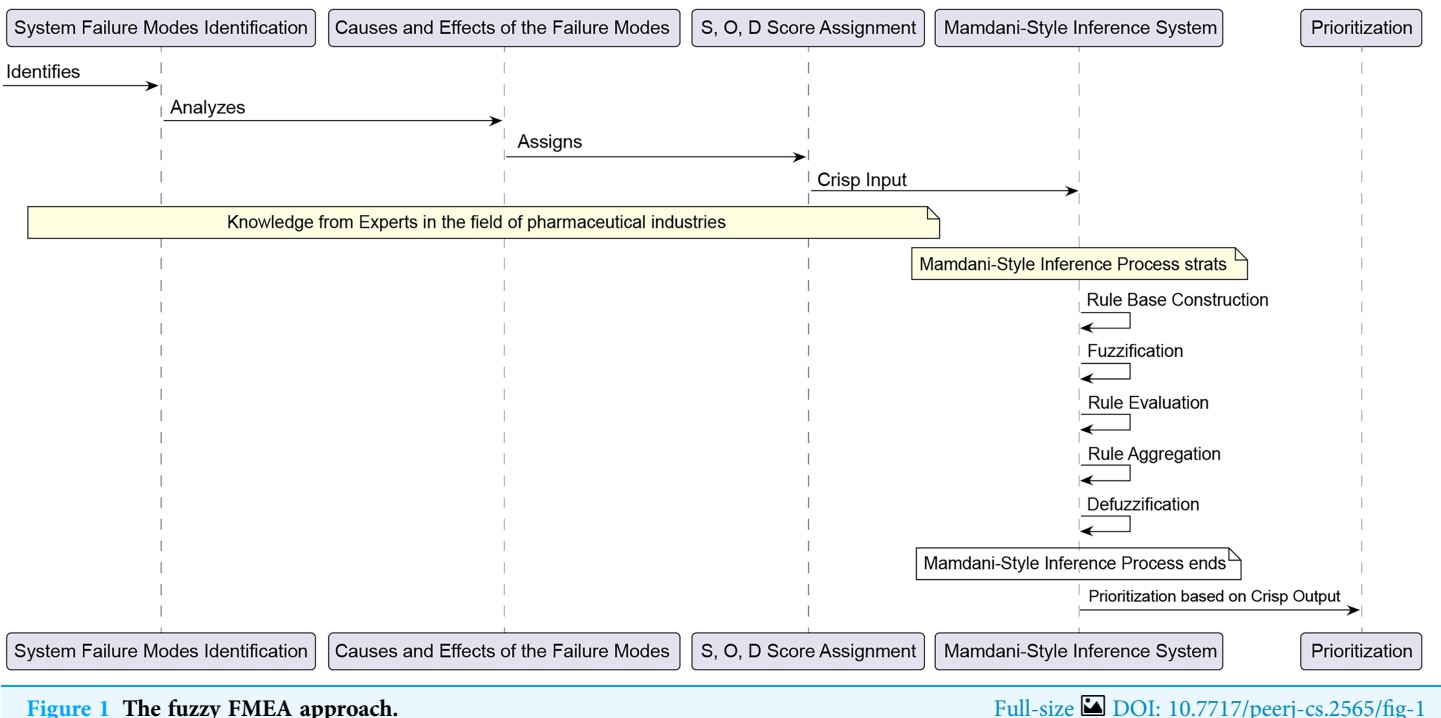

**Figure 1 The fuzzy FMEA approach.**

fuzzy sets in *O, S,* and *D*, respectively; *o, s, d,* and RPN are the input and output variables of the rule base expert system, respectively.

ii. Fuzzification

Before starting the fuzzy inference process, the crisp inputs need to be fuzzified using the membership functions to obtain an activation weight for each rule (*Akyuz, Akgun & Celik, 2016*). Thus, the second step is to transform input values into membership degree quantities, mainly ranging from 0 to 1. This process is called fuzzification.

iii. Rule Evaluation

Evaluating fuzzy rules is a critical step in the fuzzy inference process, where the activation weights for each rule are determined by assessing the membership degree of the input variables in the rule's antecedents. This step involves identifying the relevance or applicability of each rule based on the current input values. The Mamdani method applies the minimum operator as fuzzy implication for this evaluation. The output fuzzy set $R_i'$ for each rule is by Eq. (5) (*Akyuz, Akgun & Celik, 2016*):

$$\mu R_i'(RPN) = \alpha_i \wedge \mu R_i(RPN) \qquad 1 \leq i \leq K \qquad (5)$$

where $\alpha_i = \mu_{Si}(s) \wedge \mu_{Oi}(o) \wedge \mu_{Di}(d)$, representing the activation weight of each rule's antecedent.

iv. Rule Aggregation

Once the weighted contributions of each activated rule are calculated, the next step is to combine these contributions to produce the final fuzzy output. This aggregation is achieved using the maximum operator as shown in Eq. (6):

$$\mu R_i'(RPN) = \vee \mu R_i'(RPN) \qquad i = 1, 2, \ldots, K \qquad (6)$$

v. Defuzzification

Defuzzification converts fuzzy outputs from the inference process into crisp values, a necessary step when prioritizing the outputs of a fuzzy system. Several methods exist for defuzzification, including the centroid, weighted average, and bisector methods. This study used the centroid method due to its widespread application in the literature. The corresponding formula for this method is denoted as Eq. (7):

$$RPN'' = \frac{\sum\limits_{j=1}^{l} \mu Ri'(RPNj) \times RPNj}{\sum\limits_{j=1}^{l} \mu Ri'(RPNj)} \qquad (7)$$

where $RPN''$ represents the estimated value, and $l$ denotes the number of activated rules, $RPN_j$.

# APPLICATION

This section identified failure modes of the membrane-based WFI system through a fishbone diagram informed by experts' insights. A quantitative risk assessment was followed, applying the Fuzzy-FMEA method. Based on the risk assessment findings, appropriate preventive measures and process optimization recommendations have been proposed.

## Problem description

As discussed previously, microbial control is a significant challenge for a qualified membrane-based WFI system, and each critical component poses potential risks for microbial breakthrough. This study proposed a proactive approach to mitigate malfunctions in a typical membrane-based WFI system, comprising a conventional purified water generation unit and UF in the distribution loop. This system frequently triggers alarms due to microbial breakthrough issues. The following sections outline the evaluation of these failure modes using the proposed method and present strategies for preventing major defects proactively.

## Application of the proposed method

To address the main microbial hazards in membrane-based WFI systems, this study combined pharmaceutical industry expertise with relevant guidance documents. Through a cause-effect fishbone diagram and brainstorming, 24 hazards were identified across different system components. Figure 2 displays the cause-effect fishbone diagram, while Table 5 presents a detailed FMEA worksheet. This worksheet encapsulates expert insights and draws from the *Good Practice Guide: Membrane-based Water for Injection Systems*, concentrating on the potential microbial hazards inherent to these systems (*ISPE, 2022*).

Effective design and operation of membrane-based WFI systems require a thorough microbial risk assessment to identify contamination hazards. Preventive and control

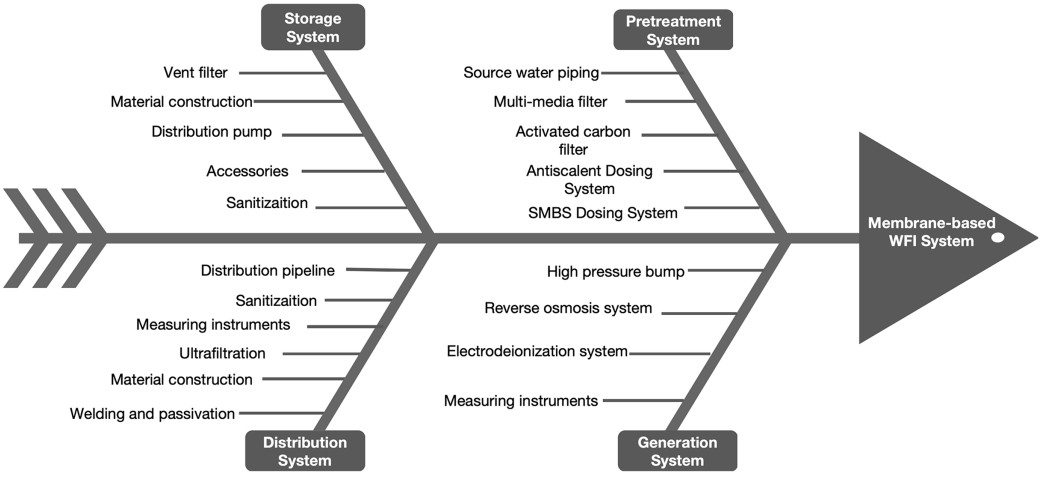

**Figure 2 The cause-effect fishbone diagram of the membrane-based WFI system.**

**Table 5 Comprehensive worksheet of membrane-based WFI system.**

| Major hazard | Code | Potential failure mode | Potential failure effect | Cause of failure |
|---|---|---|---|---|
| Pretreament system | | | | |
| Raw water transfer pipeline | FM1 | Raw water quality does not meet requirement, such as increased levels of particulates, bacteria, organic matter, minerals, *etc.* | Inefficiency of the pretreatment system in removing microbial and particulate contaminants leads to increased impurity load on downstream RO, resulting in organic and microbial contamination of the RO, decreased microbial filtration efficiency, and has potential impact on the microbial and endotoxin levels in the product water. | Water supply pipeline repairs; Rising environmental temperatures, *etc.* |
| Multi-media Filter | FM2 | Unable to filter raw water | Reduced water output, unable to meet the demand of water input of subsequent filters, affecting water production; Biofilm formed on its surface accelerates fouling of downstream RO system, decreasing microbial filtration efficiency and potentially affecting the microbial and endotoxin levels in the product water. | Delayed replacement of filter media leading to biofilm formation; Ineffective disinfection and backwashing of the filter causing clogging. |
| Activated carbon filter | FM3 | Unable to filter raw water | Inability to remove NOM leads to organic fouling of the RO membrane, affecting filtration efficiency; Nutrients provided by NOM promote uncontrolled bacterial growth on the RO membrane, leading to biofouling and potentially affecting the microbial and endotoxin levels in the product water. | Delayed replacement of filter media leading to biofilm formation; Ineffective disinfection and backwashing of the filter causing clogging. |
| Antiscalant dosing system | FM4 | Low concentration of chemicals in the antiscalant dosing tank, unable to remove precipitable ions | Insufficient antiscalant concentration leads to precipitation of ions like calcium, magnesium, and silicon on the RO, damaging the RO membrane and reducing filtration efficiency, thus potentially affecting the microbial and endotoxin levels in the product water. | Malfunction of the antiscalant dosing system |

| Major hazard | Code | Potential failure mode | Potential failure effect | Cause of failure |
|---|---|---|---|---|
| SMBS dosing system | FM5 | Low/high dosing of SMBS/SBS | Degradation of RO membrane, reduced microbial filtration efficiency, thus potentially affecting the microbial and endotoxin levels in the product water. | Malfunction of the SMBS dosing system |
| **Production system** | | | | |
| High-pressure pump | FM6 | Water Pressure may not sufficient to feed RO | The RO system is unable to remove inorganic, organic, and microbial impurities, likely affecting the microbial and endotoxin levels in the product water. | High-pressure pump malfunction |
| RO | FM7 | Leakage of the RO membrane | Causes bypass contamination, likely affecting the microbial and endotoxin levels in the product water. | Delayed maintenance of the RO; Failure of O-ring seals; Degradation and failure due to aging of the RO system. |
| | FM8 | Damage to RO membrane | Inability to remove inorganic, organic, and microbial impurities, likely affecting the microbial and endotoxin levels in the product water. | Delayed maintenance of the RO; Pressure, temperature, chemical agents, microbial metabolites, *etc.* causing membrane degradation; Improper thermal disinfection and chemical cleaning; or inadequate chlorine removal causing membrane damage. |
| | FM9 | Scaling/fouling of RO membrane | Reduced efficiency in removing inorganic, organic, and microbial impurities, potentially impacting the microbial and endotoxin levels in the product water | Frequent RO use or inadequate disinfection leading to clogging or poor feed water |
| CEDI | FM10 | CEDI cannot provide electric field | CEDI malfunction causes the anion and cation membranes and resins become a breeding ground for microorganisms, likely affecting the microbial and endotoxin levels in the product water. | CEDI malfunction |
| | FM11 | Leakage of CEDI membrane | Failure of membrane sealing leads to leakage of concentrate into the permeate water, causing product water contamination, likely affecting the microbial and endotoxin levels in the product water. | Improper pressure balance or water hammer causing membrane sealing failure in the CEDI. |
| **Storage and ambient distribution system** | | | | |
| UF | FM12 | Leakage of UF | Causes bypass contamination, directly affecting the microbial and endotoxin levels in the product water. | Delayed maintenance of the UF system; Pressure, temperature, chemical agents, microbial metabolites, *etc.* causing membrane degradation. |
| | FM13 | Damage to UF membrane | Inability to remove microorganisms and endotoxins, directly affecting the microbial and endotoxin levels in the product water. | Delayed maintenance of the UF system; Pressure, temperature, chemical agents, microbial metabolites, *etc.* causing membrane degradation; Improper chemical cleaning causing damage. |
| | FM14 | Fouling of UF | Provides an environment for microbial growth, becoming a source of microbial contamination for the distribution system, directly affecting the microbial and endotoxin levels in the product water. | UF intercepts substances becoming nutrients for viable bacteria, leading to microbial growth on the membrane surface and biofilm formation. |

(Continued)

| Major hazard | Code | Potential failure mode | Potential failure effect | Cause of failure |
|---|---|---|---|---|
| Vent filter | FM15 | Unable remove microorganism in the air | Introduction of bacteria outside to the storage tank, likely affecting the microbial and endotoxin levels in the product water. | Leakage or damage of the vent filter |
| Heat exchanger | FM16 | Insufficient sealing | Introduction of bacteria outside to the storage tank and distribution pipeline, may affecting the microbial and endotoxin levels in the product water. | Leakage of the heat exchanger |
| Distribution pipeline | FM17 | Unpolished pipeline | Biofilm formation on the pipeline surface, increased microbial count in the distribution pipeline, likely affecting the microbial and endotoxin levels in the product water. | Pipeline not polished after welding or inadequate polishing |
| Distribution pump | FM18 | Insufficient pump power | Low flow velocity in the distribution pipeline leads to microbial growth, likely affecting the microbial and endotoxin levels in the product water. | Pump malfunction |
| Pipeline welding | FM19 | Inadequate welding leading to pipeline leakage | Introduction of bacteria outside to the distribution pipeline, directly affecting the microbial and endotoxin levels in the product water. | Poor welding |
| MOC (Material of construction) | FM20 | Materials with water solubility | Leaching of materials from tanks, pipelines, *etc.*, leads to microbial growth, directly affecting the microbial and endotoxin levels in the product water, especially during system disinfection. | Tanks, pipelines, *etc.*, not using GMP-grade materials |
| Disinfection system | FM21 | Inability to sanitize the system; ineffective disinfection; infrequent disinfection | Directly affects the microbial and endotoxin levels in the product water. | Malfunction of UV system, thermal sanitation unit or ozone system |
| Auxiliary system | | | | |
| PLC (Programmable Logic Controller) system | FM22 | Critical equipment process parameters cannot be monitored. | May affect the microbial and endotoxin levels in the product water. | System malfunction |
| Measuring instruments | FM23 | Water quality cannot be measured. | May affect the microbial and endotoxin levels in the product water. | Instruments malfunction |
| Calibration instruments | FM24 | Measuring instruments cannot be calibrated. | May affect the microbial and endotoxin levels in the product water. | Instruments malfunction |

actions are necessary to prevent microbial proliferation and enhance system robustness. The FMEA worksheet provides a practical overview, listing identified hazards and analyzing the potential failure modes and causes of each. For instance, the RO system presents a major hazard, contributing to failure modes FM7, FM8, and FM9. For FM9, the effect is "reduced efficiency in removing inorganic, organic, and microbial impurities, potentially impacting the microbial and endotoxin levels in product water," caused by "frequent RO system use or inadequate disinfection leading to clogging or poor feed water quality."

Table 6 Experts evaluation for each failure mode.

| Code | Expert 1 | | | Expert 2 | | | Expert 3 | | |
|------|---|---|---|---|---|---|---|---|---|
| | O | S | D | O | S | D | O | S | D |
| FM1 | 7 | 2 | 1 | 3 | 3 | 4 | 4 | 6 | 3 |
| FM2 | 5 | 4 | 1 | 1 | 2 | 3 | 5 | 7 | 4 |
| FM3 | 6 | 5 | 1 | 3 | 5 | 3 | 5 | 7 | 4 |
| FM4 | 5 | 4 | 2 | 2 | 2 | 3 | 3 | 6 | 4 |
| FM5 | 5 | 3 | 2 | 2 | 1 | 3 | 4 | 5 | 4 |
| FM6 | 2 | 5 | 1 | 2 | 1 | 3 | 4 | 6 | 4 |
| FM7 | 4 | 6 | 3 | 4 | 5 | 4 | 3 | 7 | 5 |
| FM8 | 4 | 6 | 3 | 4 | 5 | 4 | 3 | 8 | 5 |
| FM9 | 9 | 8 | 3 | 7 | 5 | 4 | 8 | 7 | 5 |
| FM10 | 2 | 9 | 2 | 7 | 5 | 4 | 5 | 6 | 4 |
| FM11 | 2 | 9 | 3 | 4 | 5 | 7 | 4 | 7 | 5 |
| FM12 | 2 | 10 | 3 | 4 | 10 | 5 | 3 | 8 | 6 |
| FM13 | 2 | 10 | 3 | 4 | 10 | 5 | 3 | 8 | 6 |
| FM14 | 3 | 10 | 3 | 6 | 10 | 5 | 4 | 7 | 6 |
| FM15 | 2 | 9 | 7 | 4 | 10 | 3 | 3 | 7 | 4 |
| FM16 | 2 | 9 | 7 | 3 | 10 | 3 | 4 | 6 | 4 |
| FM17 | 2 | 5 | 2 | 2 | 7 | 4 | 3 | 7 | 5 |
| FM18 | 2 | 5 | 2 | 2 | 5 | 3 | 4 | 6 | 4 |
| FM19 | 5 | 10 | 2 | 2 | 8 | 2 | 3 | 7 | 5 |
| FM20 | 5 | 10 | 1 | 1 | 9 | 3 | 2 | 8 | 6 |
| FM21 | 5 | 10 | 2 | 3 | 10 | 2 | 3 | 9 | 5 |
| FM22 | 3 | 5 | 2 | 1 | 5 | 2 | 2 | 7 | 4 |
| FM23 | 5 | 10 | 2 | 3 | 5 | 2 | 2 | 7 | 4 |
| FM24 | 5 | 10 | 2 | 3 | 5 | 2 | 2 | 7 | 4 |

To evaluate the risk level of each failure mode, Fuzzy-FMEA was employed to calculate the RPN. Pharmaceutical experts rated each failure mode for severity, occurrence, and detection based on their experience and knowledge. The mean of these ratings was utilized to calculate the fRPN values in MATLAB, prioritizing them from the highest to the lowest.

In MATLAB's fuzzy logic program for FMEA, three input variables, including severity, occurrence, and detection, are defined, along with one output variable, fRPN. This study involved three senior pharmaceutical water system practitioners to evaluate each failure mode, and the results are presented in Table 6. Severity and detection use five-level triangular MFs, while occurrence employs a five-level trapezoidal MF, and fRPN has a ten-level triangular MF (Figs. 3–6).

Each input variable has five descriptors (very low, low, medium, high, and very high), totaling 15 descriptions, while the output variable fRPN includes ten levels (none, very low, low, high low, low medium, medium, high medium, low high, high, and very high). The fuzzy rule base, with 125 unique rules, is presented in Appendix 1. For instance, Rule 16
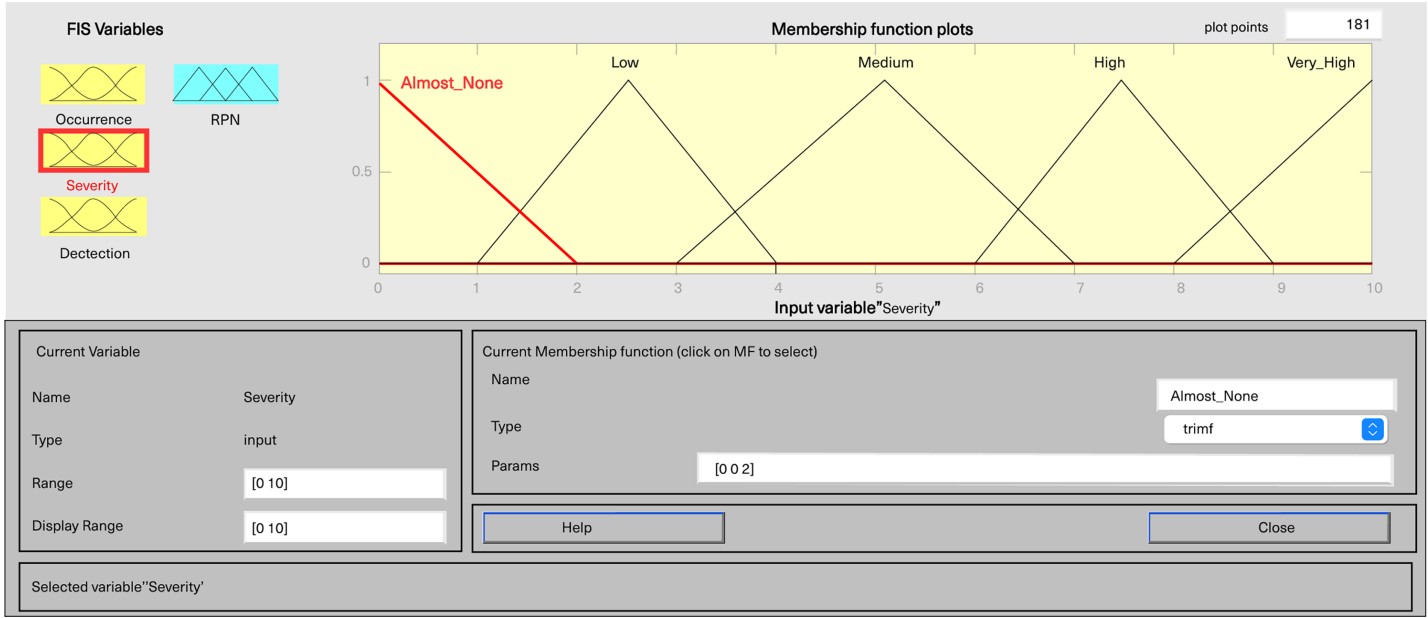

**Figure 3 Fuzzy rating for severity and their membership function.**

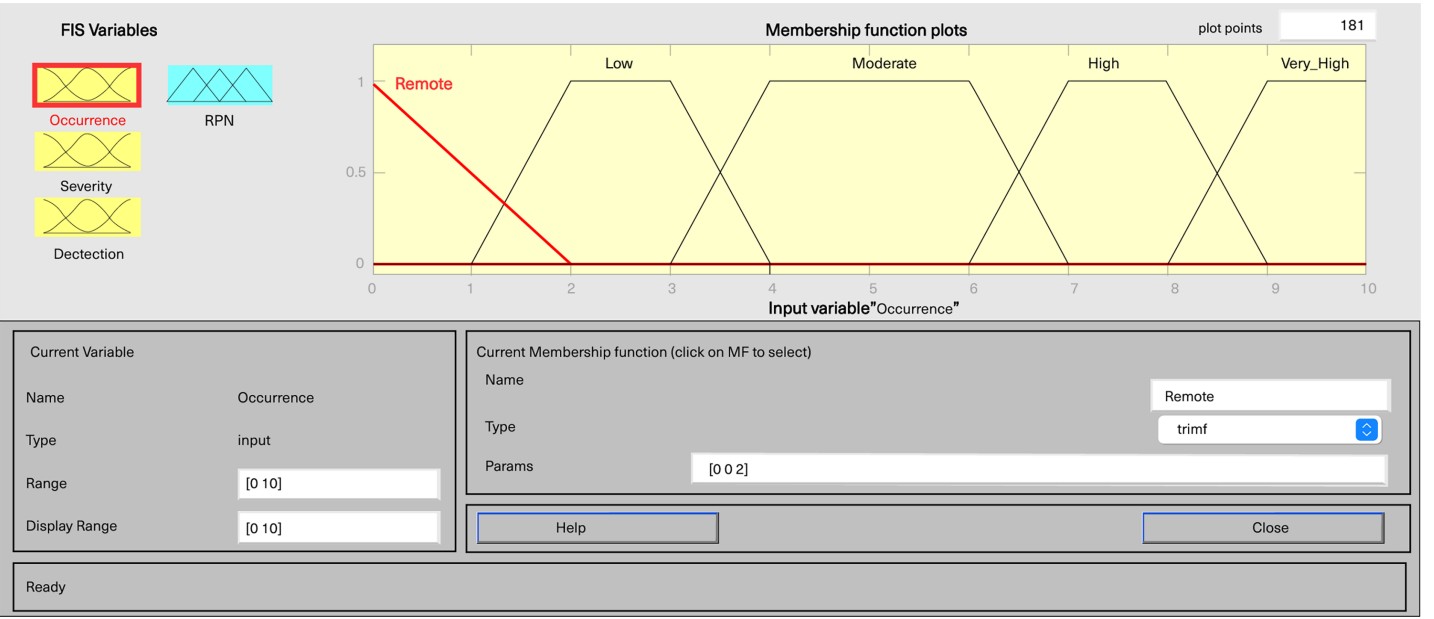

**Figure 4 Fuzzy rating for occurrence and their membership function.**

indicates that with remote occurrence, high severity, and certain detection, the RPN is low. In contrast, Rule 50 indicates that when occurrence is low, severity is very high, and detection is very low, then the resulting RPN is rated as high.

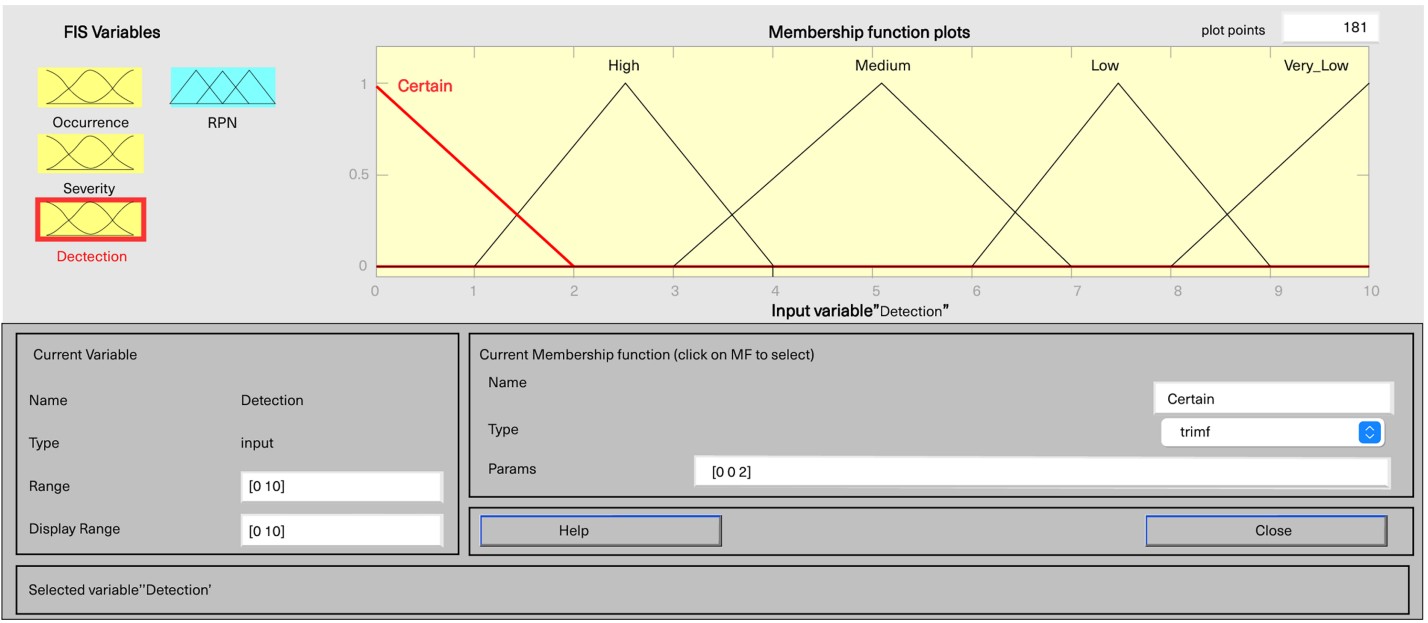

**Figure 5** Fuzzy rating for detection and their membership function.

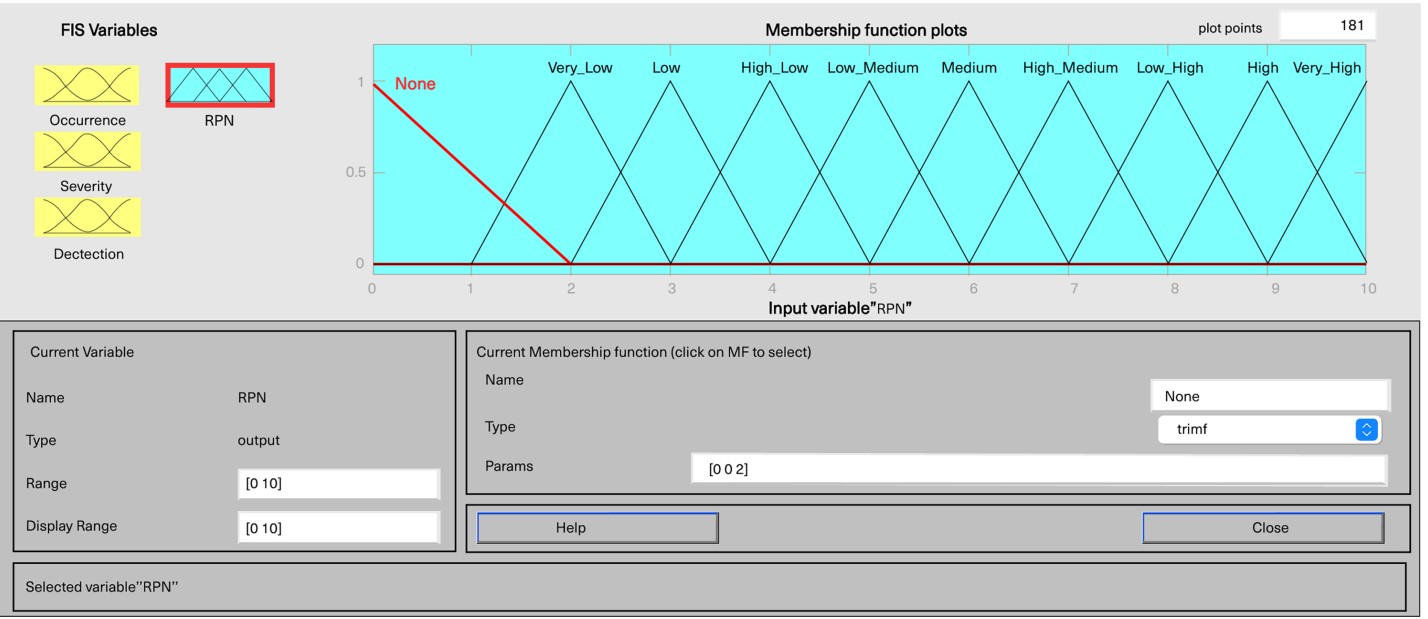

**Figure 6** Fuzzy rating for RPN and their membership function.

## Results and Discussion

The findings revealed notable differences in risk ranking between the two assessment methods. Traditional FMEA has limitations, which can lead to inefficiencies and increased costs due to excessive response measures. For example, FM11 and FM16 have received the

same RPN scores in traditional FMEA were ranked differently when assessed through Fuzzy-FMEA, providing a more informed basis for preventive measures. To mitigate this, this study utilized fRPN values to develop more scientifically based and balanced preventive and control strategies, along with optimized recommendations.

Expert evaluations indicate that the fRPN values for failure modes in both the distribution and production systems are relatively high. This is due to the close correlation between the distribution system and the drug production system, where the failure modes of each influence the other significantly. Additionally, certain failure modes in the pretreatment system, such as FM3 and FM2, also exhibit high fRPN values, highlighting their remarkable impact. The Fuzzy-FMEA method provides several practical advantages for microbial risk assessment in WFI systems. By applying fuzzy logic to severity, occurrence, and detection ratings, it can capture subtle variations in expert judgment, leading to a more refined prioritization of failure modes. This method is particularly beneficial in complex systems, such as membrane-based WFI systems, where the potential interactions between failure modes and microbial hazards are multifaceted. However, despite its benefits, Fuzzy-FMEA has limitations. Firstly, the method requires sophisticated software tools and computational resources, and the accuracy of its results heavily depends on the quality of the fuzzy rules and membership functions defined by experts. Additionally, while fuzzy logic provides flexibility, it may introduce ambiguity in cases where membership functions overlap significantly. As a result, practical application of Fuzzy-FMEA necessitates thorough validation and recalibration based on real-world system feedback to ensure reliable risk prioritization.

Table 7 demonstrates that FM14 has the highest RPN, reaching 7.00. The ultrafiltration (UF) system, serving as a microbial barrier, is typically positioned at the end of the ambient distribution system before the point of use to ensure that product water meets microbial and endotoxin standards (ISPE, 2022). However, retained substances on the UF surface can act as nutrients for viable bacteria in the distribution system, potentially leading to biofilm formation on membrane and pipeline surfaces. Consequently, improper UF system design or operation may turn it into a contamination source in the distribution system, impacting the microbial and endotoxin quality of the product water. Furthermore, if the distribution system's sterility cannot be assured, placing UF before the point of use may not suffice to meet microbial and endotoxin requirements.

FM16 is also a significant failure mode in the distribution system, with the second-highest RPN of 6.73. Any failure mode introducing external contamination can negatively affect the product water's microbial quality, as the introduction of bacteria can lead to biofilm formation in the distribution pipeline (Collentro, 2016).

The RPN values rank FM7 as the third and FM9 as the fourth. The RO system, essential in the production process, removes ions, total organic carbon (TOC), bacteria, and endotoxins, thereby helping to prevent biofilm formation (ISPE, 2022). The O-ring, installed at the pressure vessel's end-cap and in RO component connectors, prevents bypass contamination; however, insufficient sealing can compromise bacterial removal, causing bypass contamination (Fujioka et al., 2019; Liu et al., 2013; Pype et al., 2016). Additionally, membrane fouling significantly reduces RO membrane filtration

| Table 7 | Fuzzy RPN values. | | | |
|---------|-------------------|---------------|-----------|---------------|
| Code | RPN | Prioritization | Fuzzy RPN | Prioritization |
| FM14 | 182.00 | 2 | 7.00 | 1 |
| FM16 | 116.67 | 5 | 6.73 | 2 |
| FM7 | 88.00 | 9 | 6.59 | 3 |
| FM15 | 121.33 | 4 | 6.39 | 4 |
| FM9 | 213.33 | 1 | 6.39 | 4 |
| FM11 | 116.67 | 5 | 6.24 | 5 |
| FM8 | 92.89 | 8 | 6.13 | 6 |
| FM17 | 54.19 | 14 | 6.00 | 7 |
| FM3 | 70.52 | 12 | 6.00 | 7 |
| FM12 | 130.67 | 3 | 6.00 | 7 |
| FM13 | 130.67 | 3 | 6.00 | 7 |
| FM2 | 42.37 | 17 | 5.64 | 8 |
| FM21 | 106.33 | 6 | 5.64 | 8 |
| FM4 | 40.00 | 18 | 5.41 | 9 |
| FM20 | 80.00 | 11 | 5.27 | 10 |
| FM23 | 65.19 | 13 | 5.24 | 11 |
| FM24 | 65.19 | 13 | 5.24 | 11 |
| FM1 | 45.63 | 15 | 5.18 | 12 |
| FM19 | 83.33 | 10 | 5.11 | 13 |
| FM10 | 103.70 | 7 | 5.01 | 14 |
| FM22 | 30.22 | 19 | 5.00 | 15 |
| FM6 | 28.44 | 20 | 5.00 | 16 |
| FM18 | 42.67 | 16 | 5.00 | 16 |
| FM5 | 33.00 | 19 | 3.64 | 17 |

performance, impacting both production flux and filtration efficiency (*AlSawaftah et al., 2021*).

FM15 ranks fourth among all the failure modes. The vent filter on the storage tank enables air or gases to flow in and out while acting as a microbial barrier to maintain sterility. Studies indicated that the filter material in respirators could undergo oxidation and degradation after 3–6 months of exposure to high temperatures and ozone, requiring regular filter replacement (*Jornitz, Jornitz & Meltzer, 2008*). If not replaced promptly, prolonged exposure could cause the organic materials in the filter to degrade, reducing its effectiveness against microorganisms (*Tidswell & Bennett, 2017*).

Another critical failure mode, FM11, involves continuous electrodeionization (CEDI), which is typically installed downstream of the RO system for polishing deionization. CEDI also supports microbial control due to its electric field and the ionization of water into $OH^-$ and $H^+$ ions (*ISPE, 2022*). Improper pressure balance, however, can lead to membrane or seal failure, resulting in either internal leaks between concentrate and dilute channels or significant external leaks (*ISPE, 2022*). Malfunctioning CEDI can fail to reduce TOC and

inactivate microorganisms, raising the risk of biofilm formation in the distribution system, making proper CEDI operation essential.

Fouling on membrane surfaces can be managed by a well-designed pretreatment system, minimizing fouling risks (*Abushawish et al., 2023*). A key component in pretreatment is the carbon filter, which has a high fRPN ranking. The media's large surface area facilitates microbial growth, and granular activated carbon (GAC) provides approximately 35,000 $m^2/m^3$ of surface area for bacterial attachment through pores larger than 500 nm in radius (*Knezev, 2015*).

Research on microbial activity in GAC has demonstrated that fine GAC particles in filtered water could harbor bacteria concentrations 2–3 orders of magnitude higher than in the filtered water alone (*Alves et al., 2019*; *Kempisty et al., 2019*; *Lu et al., 2021*; *Oh et al., 2016*; *Tyagi et al., 2020*). Additionally, bacteria on GAC surfaces are shielded from UV radiation and chlorine disinfection due to protective adsorption sites on the carbon, limiting the effectiveness of these disinfectants (*Atabaki, Idris & Siong, 2014*; *Patel et al., 2020*). Consequently, activated carbon filters are generally not recommended in membrane-based WFI systems.

## Preventive actions and recommendations for optimization process

Given the critical importance of microbial limit compliance in WFI, it is crucial to implement preventive and control measures targeting high-risk failure modes in membrane-based WFI systems. Besides adopting preventive strategies, process optimization is equally important to reduce microbial risks and ensure compliance.

To minimize contamination, it is advisable to avoid filters with large filter surfaces, such as activated carbon filters and softeners, which are particularly susceptible to microbial buildup. Installing medium-pressure ultraviolet (UV) light before the CEDI unit can effectively inactivate bacteria, helping to prevent microbial growth in the CEDI itself.

System configuration should prioritize minimizing conditions that encourage microbial growth. This includes removing organic and inorganic matter that can act as nutrients during pre-treatment and production, limiting surfaces that facilitate microbial attachment, and ensuring nearly sterile water enters the distribution system. By implementing these measures, a membrane-based WFI system can reliably produce water meeting strict quality standards.

This study established preventive measures and process optimization strategies for membrane-based WFI systems based on pharmaceutical industry experts' insights and extensive experience. These recommendations are summarized in Appendix 2.

## CONCLUSIONS

This study provided a comprehensive microbial risk assessment for membrane-based WFI systems, concentrating on identifying failure modes that increase microbial load and implementing targeted preventive measures. By applying Fuzzy-FMEA, this research could overcome the limitations of traditional FMEA, providing a more notable, scientifically grounded evaluation of microbial risks across critical system components. Key findings indicated that failure modes in the distribution and production systems pose the highest

microbial contamination risk, especially in the UF and RO systems, which, if improperly managed, can significantly impact microbial and endotoxin levels.

The proposed application of the Fuzzy-FMEA to microbial risk assessment in membrane-based WFI systems is particularly novel and relevant for clinical specialties. This approach not only addresses a gap in existing methods, but also provides a targeted risk management framework adaptable to the high-quality demands and stringent microbial controls in healthcare and pharmaceutical industries. The novel integration of fuzzy logic in the FMEA demonstrated its utility in prioritizing risk factors, and the fRPN values revealed areas of vulnerability that were previously undervalued by traditional methods. Specifically, the UF system and components, such as vent filters and activated carbon filters emerged as high-risk areas, suggesting that preventive measures, involving reducing organic and inorganic matter, optimizing UV placement, and minimizing surface areas for microbial attachment could effectively mitigate microbial proliferation. This research highlighted valuable insights for the pharmaceutical industry, providing a robust basis for more resilient design, operation, and maintenance of membrane-based WFI systems that meet stringent quality standards. The findings and recommendations serve as a practical guide for industry stakeholders in advancing microbial risk management practices in WFI production. Future research will explore the integration of real-time monitoring tools with the Fuzzy-FMEA for continuous microbial risk assessment. Studies concentrating on advancements in microbial-resistant materials for membrane and filtration systems are beneficial for further minimizing microbial contamination risks. Additionally, developing adaptive fuzzy systems that adjust to evolving risk factors in membrane-based WFI systems could provide a dynamic approach to risk management. Lastly, applying the Fuzzy-FMEA to other water quality standards in pharmaceutical applications will broaden its utility and improve overall quality assurance practices.

### Funding
The authors received no funding for this work.

### Competing Interests
The authors declare that they have no competing interests.

### Author Contributions
- Luoyin Zhu conceived and designed the experiments, performed the experiments, analyzed the data, performed the computation work, prepared figures and/or tables, authored or reviewed drafts of the article, and approved the final draft.
- Yi Liang analyzed the data, authored or reviewed drafts of the article, and approved the final draft.

### Data Availability
   The raw measurements are available in the Supplemental File.

## Supplemental Information

Supplemental information for this article can be found online at http://dx.doi.org/10.7717/peerj-cs.2565#supplemental-information.

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
