# Peer review of "Quality risk management for microbial control in membrane-based water for injection production using fuzzy-failure mode and effects analysis"

_PeerJ Computer Science, doi:10.7717/peerj-cs.2565_

## Round 0.1 · original submission · Major Revisions

Dear authors,

Thank you for submitting your article. Reviewers have now commented on your article and suggest major revisions. We do encourage you to address the concerns and criticisms of the reviewers and resubmit your article once you have updated it accordingly. When submitting the revised version of your article, it will be better to address the following:

1. Please highlight the motivation and novelty of the proposed method considering the clinical specialties.
2. Equations should be used with correct equation number. Please do not use “as follows”, “given as”, etc. Explanation of the equations should also be checked. All variables should be written in italic as in the equations. Their definitions and boundaries should be defined. Necessary references should be provided.
3. Pros and cons of the method should be clarified. What are the limitation(s) methodology(ies) adopted in this work? Please indicate practical advantages, and discuss research limitations.
4. Future research directions should be included.

Best wishes,

Reviewer 1 ·

Basic reporting

This study evaluates the potential microbial risks in water for injection (WFI) systems that utilize membrane-based processes. Using a Fuzzy Failure Mode and Effects Analysis (Fuzzy-FMEA) approach, the research identifies and prioritizes failure modes based on the Risk Priority Number (RPN). The study recommends preventive measures to control failure modes that could increase the microbial load and mitigate their impact. The application of fuzzy FMEA in the microbial risk assessment of these systems is proposed for the first time, offering a theoretical contribution to the field. Overall, it's well-written. More information is needed in the introduction section. It is suggested that the authors provide more justification for their study, specifically by expanding upon the knowledge gap being filled. The second section literature review can be combined with introduction.

Experimental design

no comment

Validity of the findings

How does this method compare with other commonly used method? Any data to support the comparison?

·

Basic reporting

The research article entitled “Quality risk management for microbial control of membrane based WFI systems using Fuzzy-FMEA” is interesting and can contribute well to the scientific field. There are recommendations for the authors to make the manuscript more effective for readers. The manuscript can be considered for publication on addressing the following minor comments.
1. The authors have to improve the language of the manuscript for clarity and professionalism
2. Justify the paragraphs, distribute the texts evenly between the paragraphs and make it more polished
3. The scientific names have to be in italics. For example: Legionella pneumophila (84), Cryptosporidium (92) italicize them
4. Avoid unnecessary capitalization of each word throughout the manuscript. For example : An Integrated, Methodology (110) use lowercase
5. Make sure that all the abbreviations are expanded once throughout the manuscript. For example : AHP and GTST expanded forms are missing
6. The format of the manuscript is not consistent and the grammar as well as the tense consistency is not maintained. Make it clearer
7. Make sure all references are formatted the same way. Double check the format. Try to use recent articles. For example : Bancroft 365 et al., 1983; Cairo, McElhaney & Suffet, 1979; Chudyk & Snoeyink, 1984; Servais, Cauchi & 366 Billen, 1994; Van der Kooij, 1983; Weber, Pirbazari & Melson, 1978

8. Try to improve the clarity of sentences to avoid confusions. For example : “it is a quality control tool designed to pinpoint potential factors causing an overall effect and prevent product quality defects” (132)

Experimental design

1. Check overall flow of the manuscript. The transition from one section to the other could be smoother
2. It feels like a report rather than a research article. The authors should rephrase the tone of the manuscript to make it a bit more professional. It is requested to consult a native English speaker for enhancing the tone of the manuscript

Validity of the findings

No Comment

Additional comments

No Comment

Reviewer 3 ·

Basic reporting

The article presents notable findings, introducing Fuzzy Failure Mode and Effects Analysis for microbial risk assessment in membrane-based Water for Injection systems, especially cold WFI production. It identifies high-risk failure modes such as UF membrane fouling, insufficient heat exchanger sealing, RO membrane leakage, and vent filter issues, offering process optimization and preventive measures. The study demonstrates Fuzzy-FMEA's superiority over traditional FMEA and its first application in WFI microbial risk assessment, providing valuable theoretical and practical insights. These are strong points of the article. However, the following points need to be addressed before the article can be considered for publication.
1. Avoid using uncommon abbreviations in the title of the article.
2. The title and keywords section should not contain repetitive terms.
3. Numerous punctuation, grammatical, and syntax errors were identified throughout the manuscript. It is recommended that these issues be carefully addressed before resubmission.
4. The abstract should be written as a single, cohesive paragraph.
5. Instead of splitting the introduction into multiple short paragraphs, it is advised to limit it to two or three well-structured paragraphs.
6. The last paragraph of the introduction (Lines 60-66) does not add any value to the overall quality of the manuscript.
7. The literature review section should be merged with the introduction to reduce the overall length. The content from the literature review can be used to highlight the knowledge gap in the introduction and to support findings in the discussion section.
8. The resolution of all figures must be improved to meet publication standards.
9. Relevant references should be cited for Table 5.
10. Many of the references used are outdated and do not reflect recent advancements in the field. These should be replaced with more current literature from the last five years.
11. The conclusion is underdeveloped and repeats generic information (L 401-403) from the abstract and introduction. It should be rewritten to emphasize key findings.
12. It is inappropriate for an author to be acknowledged in the acknowledgments section of the same article. Additionally, the sentence structure in this section requires significant revision.

Experimental design

no comment

Validity of the findings

no comment

---

## Round 0.2 · accepted · Accept

Dear Authors,

I am grateful for your efforts in revising the paper. Two of the previous reviewers did not agree to re-review. However, the other reviewer has indicated that your revised paper is suitable for acceptance in its current form. I am also satisfied with the revised manuscript and believe it is now ready for publication.

Best wishes,

·

Basic reporting

All comments raised was addressed and incorporated in the revised manuscript

Experimental design

All comments raised was addressed and incorporated in the revised manuscript

Validity of the findings

All comments raised was addressed and incorporated in the revised manuscript

Additional comments

All comments raised was addressed and incorporated in the revised manuscript